# Genome-wide association and Mendelian randomisation analysis provide insights into the pathogenesis of heart failure

Sonia Shah ⓘ et al.[#]

Heart failure (HF) is a leading cause of morbidity and mortality worldwide. A small proportion of HF cases are attributable to monogenic cardiomyopathies and existing genome-wide association studies (GWAS) have yielded only limited insights, leaving the observed heritability of HF largely unexplained. We report results from a GWAS meta-analysis of HF comprising 47,309 cases and 930,014 controls. Twelve independent variants at 11 genomic loci are associated with HF, all of which demonstrate one or more associations with coronary artery disease (CAD), atrial fibrillation, or reduced left ventricular function, suggesting shared genetic aetiology. Functional analysis of non-CAD-associated loci implicate genes involved in cardiac development (*MYOZ1*, *SYNPO2L*), protein homoeostasis (*BAG3*), and cellular senescence (*CDKN1A*). Mendelian randomisation analysis supports causal roles for several HF risk factors, and demonstrates CAD-independent effects for atrial fibrillation, body mass index, and hypertension. These findings extend our knowledge of the pathways underlying HF and may inform new therapeutic strategies.

[#]A full list of authors and their affiliations appears at the end of the paper.

Heart failure (HF) affects >30 million individuals worldwide and its prevalence is rising[1]. HF-associated morbidity and mortality remain high despite therapeutic advances, with 5-year survival averaging ~50%[2]. HF is a clinical syndrome defined by fluid congestion and exercise intolerance due to cardiac dysfunction[3]. HF results typically from myocardial disease with impairment of left ventricular (LV) function manifesting with either reduced or preserved ejection fraction. Several cardiovascular and systemic disorders are implicated as aetiological factors, most notably coronary artery disease (CAD), obesity and hypertension; multiple risk factors frequently co-occur and the contribution to aetiology has been challenging based on observational data alone[1,4]. Monogenic hypertrophic and dilated cardiomyopathy (DCM) syndromes are known causes of HF, although they account for a small proportion of disease burden[5]. HF is a complex disorder with an estimated heritability of ~26%[6]. Previous modest-sized genome-wide association studies (GWAS) of HF reported two loci, while studies of DCM have identified a few replicated loci[7–11]. We hypothesised that a GWAS of HF with greater power would provide an opportunity for: (i) discovery of genetic variants modifying disease susceptibility in a range of comorbid contexts, both through subtype-specific and shared pathophysiological mechanisms, such as fluid congestion; and (ii) provide insights into aetiology by estimating the unconfounded causal contribution of observationally associated risk factors by Mendelian randomisation (MR) analysis[12].

Herein, we perform a large meta-analysis of GWAS of HF to identify disease associated genomic loci. We seek to relate HF-associated loci to putative effector genes through integrated analysis of expression data from disease-relevant tissues, including statistical colocalisation analysis. We evaluate the genetic evidence supporting a causal role for HF risk factors identified through observational studies using Mendelian randomisation and explore mediation of risk through conditional analysis. In summary, our study identifies additional HF risk variants, prioritises putative effector genes and provides a genetic appraisal of the putative causal role of observationally associated risk factors, contributing to our understanding of the pathophysiological basis of HF.

## Results

**Meta-analysis identifies 11 genomic loci associated with HF.** We conducted a GWAS comprising 47,309 cases and 930,014 controls of European ancestry across 26 studies from the Heart Failure Molecular Epidemiology for Therapeutic Targets (HERMES) Consortium. The study sample comprised both population cohorts (17 studies, 38,780 HF cases, 893,657 controls) and case-control samples (9 studies, 8,529 cases, 36,357 controls; see Supplementary Notes 2 and 3 for a detailed description of the included studies). Genotype data were imputed to either the 1000 Genomes Project (60%), Haplotype Reference Consortium (35%) or study-specific reference panels (5%). We performed a fixed-effect inverse variance-weighted (IVW) meta-analysis relating 8,281,262 common and low-frequency variants (minor allele frequency (MAF) > 1%) to HF risk (Fig. 1). We identified 12 independent genetic variants, at 11 loci associated with HF at genome-wide significance ($P < 5 \times 10^{-8}$), including 10 loci not previously reported for HF (Fig. 2, Table 1). The quantile–quantile, regional association plots and study-specific effects for each independent variant are shown in Supplementary Figs. 1–3. We replicated two previously reported associations for HF and three of four loci for DCM (Bonferroni-corrected $P < 0.05$; Supplementary Data 1). Using linkage disequilibrium score regression (LDSC)[13], we estimated the heritability of HF in UK

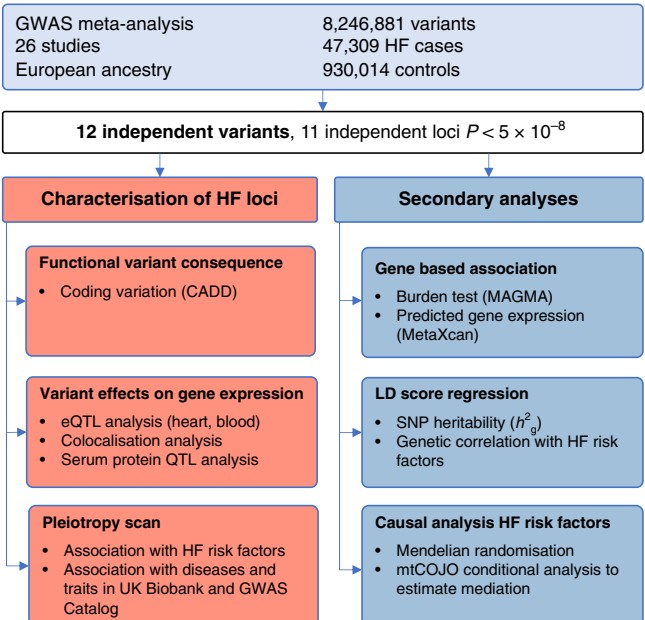

**Fig. 1 Study design and analysis workflow.** Overview of study design to identify and characterise heart failure-associated risk loci and for secondary cross-trait genome-wide analyses. GWAS, genome-wide association study; QTL, quantitative trait locus; MAGMA, Multi-marker Analysis of GenoMic Annotation; SNP, single-nucleotide polymorphism; mtCOJO, multi-trait-based conditional and joint analysis.

Biobank ($h_g^2$) on the liability scale, as 0.088 (s.e. = 0.013), based on an estimated disease prevalence of 2.5%[14].

**Phenotypic effects of HF-associated variants.** Next, we investigated associations between the identified loci and other traits that may provide insights into aetiology. First, we queried the NHGRI-EBI GWAS Catalog[15] and a large database of genetic associations in UK Biobank (http://www.nealelab.is/uk-biobank), and identified several biomarker and disease associations at each locus (Supplementary Data 2 and 3). Second, we tested for associations of identified loci with ten known HF risk factors, including cardiac structure and function measures, using GWAS summary data (Supplementary Data 4)[16–23]. Six sentinel variants were associated with CAD, including established loci, such as 9p21/*CDKN2B-AS1* and *LPA*[18]. Four variants were associated with atrial fibrillation (AF), a common antecedent and sequela of HF[24]. To estimate whether the HF risk effects were mediated wholly or in part by risk factors upstream of HF (e.g., CAD), we conditioned HF GWAS summary statistics on nine HF risk factors using Multi-trait Conditional and Joint Analysis (mtCOJO)[25] (Supplementary Data 5). Conditioning on AF attenuated the HF risk effect by >50% for the *PITX2/FAM241A* locus but not other AF-associated loci (*KLHL3, SYNPOL2/AGAP5*), conditioning on CAD fully attenuated effects for two of the six CAD loci (*LPA*, 9p21/*CDKN2B-AS1*) and conditioning on body mass index (BMI) ablated the effect of the *FTO* locus (Supplementary Fig. 4, Supplementary Data 5). Next, we performed hierarchical agglomerative clustering of loci based on cross-trait associations to identify groups related to HF subtypes (Fig. 3). Among HF loci not associated with CAD, a group of four clustered together, of which two (*KLHL3* and *SYNPO2L/AGAP5*) were associated with AF and two (*BAG3* and *CDKN1A*) with reduced LV systolic function (fractional shortening (FS); Bonferroni-corrected $P < 0.05$); we highlight the results for these loci in our reporting of subsequent analyses to identify candidate genes. Notably, genetic

---

The figure (Fig. 1) contains the following boxes:

GWAS meta-analysis — 8,246,881 variants
26 studies — 47,309 HF cases
European ancestry — 930,014 controls

**12 independent variants**, 11 independent loci $P < 5 \times 10^{-8}$

**Characterisation of HF loci**

**Functional variant consequence**
- Coding variation (CADD)

**Variant effects on gene expression**
- eQTL analysis (heart, blood)
- Colocalisation analysis
- Serum protein QTL analysis

**Pleiotropy scan**
- Association with HF risk factors
- Association with diseases and traits in UK Biobank and GWAS Catalog

**Secondary analyses**

**Gene based association**
- Burden test (MAGMA)
- Predicted gene expression (MetaXcan)

**LD score regression**
- SNP heritability ($h_g^2$)
- Genetic correlation with HF risk factors

**Causal analysis HF risk factors**
- Mendelian randomisation
- mtCOJO conditional analysis to estimate mediation

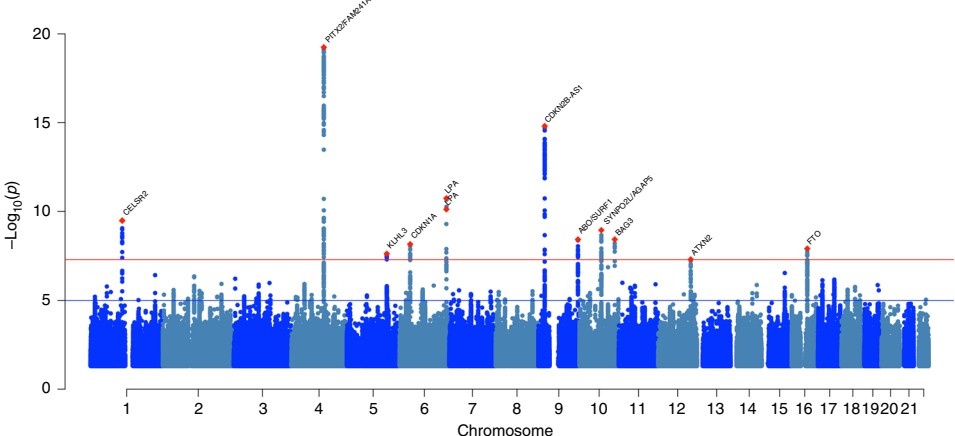

**Fig. 2 Manhattan plot of genome-wide heart failure associations.** The *x*-axis represents the genome in physical order; the *y*-axis shows $-\log_{10} P$ values for individual variant association with heart failure risk from the meta-analysis ($n = 977,323$). Suggestive associations at a significance level of $P < 1 \times 10^{-5}$ are indicated by the blue line, while genome-wide significance at $P < 5 \times 10^{-8}$ is indicated by the red line. Meta-analysis was performed using a fixed-effect inverse variance-weighted model. Independent genome-wide significant variants are annotated with the nearest gene(s).

associations with DCM at the *BAG3* locus have been reported previously[10,11].

**Tissue-enrichment analysis**. We performed gene-based association analyses using MAGMA[26] to identify tissues and aetiological pathways relevant to HF. Thirteen genes were associated with HF at genome-wide significance, of which four were located within 1 Mb of a sentinel HF variant and expressed in heart tissue (Supplementary Data 6). Tissue specificity analysis across 53 tissue types from the Genotype-Tissue Expression (GTEx) project identified the atrial appendage as the highest ranked tissue for gene expression enrichment, excluding reproductive organs (Supplementary Fig. 5). We sought to map candidate genes to the HF loci by assessing the functional consequences of sentinel variants (or their proxies) on gene expression, and protein structure/abundance using quantitative trait locus (QTL) analyses.

**Variant effects on protein coding sequence**. Since the identified HF variants were located in non-coding regions, we investigated if sentinel variants were in linkage disequilibrium (LD, $r^2 > 0.8$) with non-synonymous variants with predicted deleterious effects. We identified a missense variant in *BAG3* (rs2234962; $r^2 = 0.99$ with sentinel variant rs17617337) associated previously with DCM and progression to HF, and three missense variants in *SYNPO2L* (rs34163229, rs3812629 and rs60632610; all $r^2 > 0.9$ with sentinel variant rs4746140)[10,11,27]. All four missense variants had Combined Annotation Dependent Depletion scores > 20, suggesting deleterious effects (Supplementary Data 7).

**Prioritisation of putative effector genes by expression analysis**. We then sought to identify candidate genes for HF risk loci by assessing their effects on gene expression. Given that cardiac dysfunction defines HF and that HF-associated genes by MAGMA analysis were enriched in heart tissues, we first looked for expression quantitative trait loci (eQTL) in heart tissues (LV, left atrium, and RAA, right atrium auricular region) from the Myocardial Applied Genomics Network (MAGNet) and GTEx projects. Three of 12 variants were significantly associated with the expression of one or more genes located in *cis* in at least one heart tissue (Bonferroni-corrected $P < 0.05$; Supplementary Data 8). For several of the identified HF loci, extra-cardiac tissues are likely to be relevant; for example, liver is reported to mediate

effects of the LPA locus[28]. To further explore these effects, we then analysed results from a large whole-blood eQTL dataset ($n = 31,684$) and found associations with *cis*-gene expression ($P < 5 \times 10^{-8}$) for 8 of 12 sentinel variants (Supplementary Table 1)[29]. For most HF variants, heart eQTL associations were consistent with those for blood traits; however, for intronic HF sentinel variants in *BAG3*, *CDKN1A* and *KLHL3* we detected expression of the corresponding gene transcripts in blood only.

Next, to prioritise among candidate genes identified through eQTL associations, we estimated the posterior probability for a common causal variant underlying associations with gene expression and HF at each locus, by conducting pairwise Bayesian colocalisation analysis[30]. We found evidence for colocalisation (posterior probability > 0.7) for *MYOZ1* and *SYNPO2L* in heart, *PSRC1* and *ABO* in heart and blood; and *CDKN1A* in blood (Supplementary Data 8, Supplementary Table 1). *PSRC1* and *MYOZ1* were also implicated in a transcriptome-wide association analysis performed using predicted gene expression based on GTEx human atrial and ventricular expression reference data (Supplementary Table 2). Using serum pQTL data from the INTERVAL study ($N = 3,301$), we also identified significant concordant *cis* associations for *BAG3* and *ABO* (Supplementary Data 9)[31].

The evidence linking candidate genes with HF risk loci is summarised in Supplementary Table 3, and candidate genes are described in Supplementary Note 1. At HF risk loci associated with reduced systolic function or AF, but not with CAD, the annotated functions of candidate genes related to myocardial disease processes, and traits that may influence clinical expressivity, such as renal sodium handling. For example, the sentinel variant at the *SYNPO2L/AGAP5* locus was associated with expression of *MYOZ1* and *SYNPO2L*, encoding two α-actinin binding *Z*-disc cardiac proteins. *MYOZ1* is a negative regulator of calcineurin signalling, a pathway linked to pathological hypertrophy[32,33] and *SYNPO2L* is implicated in cardiac development and sarcomere maintenance[34]. The HF sentinel variant at the *BAG3* locus was in high LD with a non-synonymous variant associated previously with DCM[11], and was associated with decreased *cis*-gene expression in blood. *BAG3* encodes a *Z*-disc-associated protein that mediates selective macroautophagy and promotes cell survival through interaction with apoptosis regulator *BCL2*[35]. *CDKN1A* encodes p21, a potent cell cycle inhibitor that mediates post-natal cardiomyocyte cell cycle arrest[36] and is implicated in *LMNA*-mediated cellular stress

**Table 1 Variants associated with heart failure at genome-wide significance.**

| rsID | Chr | Position (hg19) | Nearest gene(s)[a] | Function | Risk/ref allele | RAF (%) | OR (95% CI) | P value | $I^2_{HET}$ | $P_{HET}$ |
|---|---|---|---|---|---|---|---|---|---|---|
| rs660240 | 1 | 109817838 | CELSR2 | UTR3 | C/T | 0.79 | 1.06 (1.04–1.08) | 3.25E-10 | 0 | 0.513 |
| rs17042102 | 4 | 111668626 | PITX2, FAM241A | Intergenic | A/G | 0.12 | 1.12 (1.09–1.14) | 5.71E-20 | 43.1 | 0.008 |
| rs11745324 | 5 | 137012171 | KLHL3 | Intronic | G/A | 0.77 | 1.05 (1.03–1.07) | 2.35E-08 | 5.7 | 0.381 |
| rs4135240 | 6 | 36647680 | CDKN1A | Intronic | T/C | 0.66 | 1.05 (1.03–1.07) | 6.84E-09 | 43.8 | 0.009 |
| rs55730499 | 6 | 161005610 | LPA | Intronic | T/C | 0.07 | 1.11 (1.08–1.14) | 1.83E-11 | 21.1 | 0.164 |
| rs140570886 | 6 | 161013013 | LPA | Intronic | C/T | 0.02 | 1.24 (1.16–1.3) | 7.69E-11 | 24.8 | 0.133 |
| rs1556516 | 9 | 22100176 | 9p21/CDKN2B-AS1 | ncRNA | C/G | 0.48 | 1.06 (1.05–1.08) | 1.57E-15 | 12.8 | 0.269 |
| rs600038 | 9 | 136151806 | ABO, SURF1 | Intergenic | C/T | 0.21 | 1.06 (1.04–1.08) | 3.68E-09 | 0 | 0.729 |
| rs4746140 | 10 | 75417249 | SYNPO2L, AGAP5 | Intergenic | G/C | 0.85 | 1.07 (1.05–1.09) | 1.10E-09 | 9.7 | 0.319 |
| rs17617337 | 10 | 121426884 | BAG3 | Intronic | C/T | 0.78 | 1.06 (1.04–1.08) | 3.65E-09 | 55 | 2.1E-4 |
| rs4766578 | 12 | 111904371 | ATXN2 | Intronic | T/A | 0.47 | 1.04 (1.03–1.06) | 4.90E-08 | 10.6 | 0.308 |
| rs56094641 | 16 | 53806453 | FTO | Intronic | G/A | 0.42 | 1.05 (1.03–1.06) | 1.21E-08 | 17.4 | 0.215 |

The table shows the 12 independent variants associated with HF at the genome-wide significance level ($P < 5 \times 10^{-8}$) in the meta-analysis of 29 studies. Meta-analyses were carried out using an IVW fixed-effect approach. The $I^2_{HET}$ describes the percentage of variation across the 29 studies that is due to heterogeneity. $P_{HET}$ was derived from a Cochran's Q-test (two-sided) for heterogeneity
Chr, chromosome; ncRNA, non-coding RNA; ref, reference; RAF, risk allele frequency; OR, odds ratio; CI, confidence intervals; HET, heterogeneity; $I^2$, I-squared
[a]Nearest gene with a functional protein or RNA (e.g., anti-sense RNA) product that either overlaps with the sentinel variant, or for intergenic variants, the nearest genes up- and downstream, respectively (separated by comma)

responses[37]. *KLHL3* is a negative regulator of the thiazide-sensitive Na$^+$Cl$^-$ cotransporter (*SLC12A3*) in the distal nephron; loss of function variants cause familial hyperkalaemic hypertension (FHHt) by increasing constitutive sodium and chloride resorption[38]. The sentinel variant at this locus was associated with decreased gene expression and could predispose to sodium and fluid retention. Notably, thiazide diuretics inhibit *SLC12A3* to restore sodium and potassium homoeostasis in FHHt and are effective treatments for preventing hypertensive HF[39].

**Genetic appraisal of HF risk factors.** Although many risk factors are associated with HF, only myocardial infarction and hypertension have an established causal role based on evidence from randomised controlled trials (RCTs)[40]. Important questions remain about causality for other risk factors. For instance, type 2 diabetes (T2D) is a risk factor for HF, yet it is unclear if the association is mediated via CAD risk or by direct myocardial effects, which may have important preventative implications[41]. Accordingly, we investigated potential causal roles for modifiable HF risk factors, using GWAS summary data. First, we estimated the genetic correlation ($r_g$) between HF and 11 related traits, using bivariate LDSC. For eight of the eleven traits tested, we found evidence of shared additive genetic effects with estimates of $r_g$ ranging from −0.25 to 0.67 (Supplementary Table 4). The estimated CAD-HF $r_g$ was 0.67, suggesting 45% ($r_g^2$) of variation in genetic risk of HF is accounted for by common genetic variation shared with CAD, and that the remaining genetic variation is independent of CAD.

Next, we estimated the causal effects of the 11 HF risk factors using Generalised Summary-data-based Mendelian Randomisation, which accounts for pleiotropy by excluding heterogenous variants based on the heterogeneity in dependent instrument (HEIDI) test (Methods, Supplementary Fig. 6, Supplementary Data 10). Consistent with evidence from RCTs and genetic studies[42], we found evidence for causal effects of higher diastolic blood pressure (DBP; OR = 1.30 per 10 mmHg, $P = 9.13 \times 10^{-21}$) and systolic blood pressure (SBP; OR = 1.18 per 10 mmHg, $P = 4.8 \times 10^{-23}$), and higher risk of CAD (OR = 1.36, $P = 1.67 \times 10^{-70}$) on HF. We note that the effect estimates for variant associations with blood pressure, included as instrumental variables, were adjusted for BMI, which may attenuate the estimated causal effect on HF. We found a s.d. increment of BMI (equivalent to 4.4 kg m$^{-2}$ (men) − 5.4 kg m$^{-2}$ (women)[43]) accounted for a 74% higher HF risk ($P = 2.67 \times 10^{-50}$), consistent with previous reports[44,45]. We identified evidence supporting causal effects of genetic liability to AF (OR of HF per 1 log odds higher AF = 1.19, $P = 1.40 \times 10^{-75}$) and T2D (OR of HF per 1 log odds higher T2D = 1.05, $P = 6.35 \times 10^{-05}$) and risk of HF. We did not find supportive evidence for a causal role for higher heart rate (HR) or lower glomerular filtration rate (GFR) despite reported observational associations[46,47]. We then performed a sensitivity analysis to explore potential bias arising from the inclusion of case-control samples by repeating the Mendelian randomisation analysis, using HF GWAS estimates generated from population-based cohort studies only. The results of this analysis were consistent with those generated from the overall sample (Supplementary Table 5).

To investigate whether risk factor effects on HF were mediated by CAD and AF, we performed analyses conditioning for CAD and AF using mtCOJO. We observed attenuation of the effect of T2D after conditioning for CAD (OR = 1.02, P = 0.19), suggesting at least partial mediation by CAD risk rather than through direct myocardial effects of hyperglycaemia. Similarly, the effects of low-density lipoprotein cholesterol (LDL-C) were fully explained by effects of CAD on HF risk (OR = 1.00, P = 0.80).

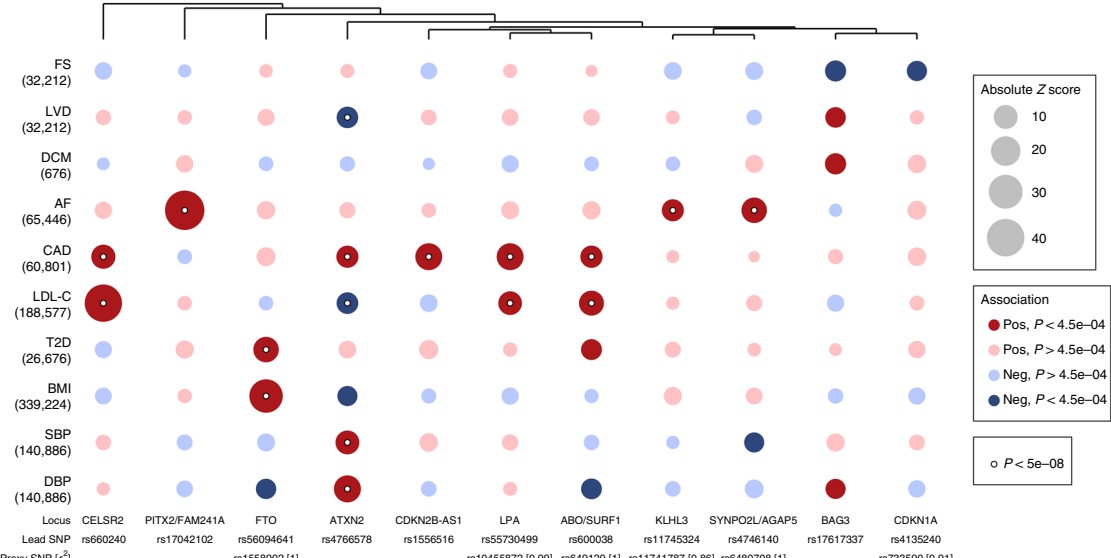

**Fig. 3 Associations of HF risk variants with traits relating to disease subtypes and risk factors.** This bubble plot shows associations between the identified HF loci and risk factors and quantitative imaging traits, using summary estimates from UK Biobank (DCM, dilated cardiomyopathy) and published GWAS summary statistics. Number in bracket represents sample size (for quantitative traits) or number of cases (for binary traits) used to derive the GWAS summary statistics. The size of the bubble represents the absolute Z-score for each trait, with the direction oriented towards the HF risk allele. Red/blue indicates a positive/negative cross-trait association (i.e., increase/decrease in disease risk or increase/decrease in continuous trait). We accounted for family-wise error rate at 0.05 by Bonferroni correction for the ten traits tested per HF locus ($P < 4.5e-4$); traits meeting this threshold of significance for association are indicated by dark colour shading. Agglomerative hierarchical clustering of variants was performed using the complete linkage method, based on Euclidian distance. Where a sentinel variant was not available for all traits, a common proxy was selected (bold text). For the LPA locus, associations for the more common of the two variants at this locus are shown. Bold text represents variants whose estimates are plotted, upon which we performed hierarchical agglomerative clustering using the complete linkage method based on Euclidian distance. FS, fractional shortening; LVD, left ventricular dimension; DCM, dilated cardiomyopathy; AF, atrial fibrillation; CAD, coronary artery disease; LDL-C, low-density lipoprotein cholesterol; T2D, type 2 diabetes; BMI, body mass index; SBP, systolic blood pressure; DBP, diastolic blood pressure.

Conversely, the effects of blood pressure, BMI and triglycerides (TGs) were only partially attenuated, suggesting causal mechanisms independent of those associated with AF and CAD (Fig. 4, Supplementary Data 10).

## Discussion

We identify 12 independent variant associations for HF risk at 11 genomic loci by leveraging genome-wide data on 47,309 cases and 930,014 controls, including 10 loci not previously associated with HF. The identified loci were associated with modifiable risk factors and traits related to LV structure and function, and include the strongest associations signals from GWAS of CAD (9p21, *LPA*)[18], AF (*PITX2*)[17] and BMI (*FTO*)[20]. Conditioning for CAD, AF and blood pressure traits demonstrated that the effects of some loci (e.g., 9p21/*CDKN2B-AS1*) were mediated wholly via risk factor trait associations (e.g., CAD); however, for 8 of 12 variants the attenuation of effects was <50%, suggesting alternative mechanisms may be important. Those loci associated with reduced LV systolic function or AF mapped to candidate genes implicated in processes of cardiac development, protein homeostasis and cellular senescence. We use genetic causal inference and conditional analysis to explore the syndromic heterogeneity and causal biology of HF, and to provide insights into aetiology. Mendelian randomisation analysis confirms previously reported casual effects for BMI and provides evidence supporting the causal role of several observationally linked risk factors, including AF, elevated blood pressure (DBP and SBP), LDL-C, CAD, TGs and T2D. Using conditional analysis, we demonstrate CAD-independent effects for AF, BMI, blood pressure and estimate that the effects of T2D are mostly mediated by an increased risk of CAD.

The heterogeneity of aetiology and clinical manifestation of HF are likely to have reduced statistical power. We identify a modest number of genetic associations for HF compared to other cardiovascular disease GWAS of comparable sample size, such as for AF, suggesting that an important component of HF heritability may be more attributable to specific disease subtypes than components of a final common pathway[17]. Subsequent studies will explore emerging opportunities to define HF subtypes and longitudinal phenotypes in large biobanks and patient registries at scale using standardised definitions based on diagnostic codes, imaging and electronic health records. We speculate that future analysis of HF subtypes may yield additional insights into the genetic architecture of HF to inform new approaches to prevention and treatment.

## Methods

**Samples.** Participants of European ancestry from 26 cohorts (with a total of 29 distinct datasets) with either a case-control or population-based study design were included in the meta-analysis, as part of the HERMES Consortium. Cases included participants with a clinical diagnosis of HF of any aetiology with no inclusion criteria based on LV ejection fraction; controls were participants without HF. Definitions used to adjudicate HF status within each study are detailed in the Supplementary Data 11 and baseline characteristics for each study are provided in Supplementary Data 12. We meta-analysed data from a total of 47,309 cases and 930,014 controls. All included studies were ethically approved by local institutional review boards and all participants provided written informed consent. The meta-analysis of summary-level GWAS estimates from participating studies was performed in accordance with guidelines for study procedures provided by the UCL Research Ethics Committee.

**Genotyping and imputation.** All studies used high-density genotyping arrays and performed genotype calling and pre-imputation quality control (QC), as reported in Supplementary Data 13. Studies performed imputation using one or more of the following reference panels: 1000 Genomes (Phase 1 or Phase 3)[48], Hapmap 2 NCBI build 36[49], Haplotype Reference Consortium (HRC)[50], the Estonian Whole-

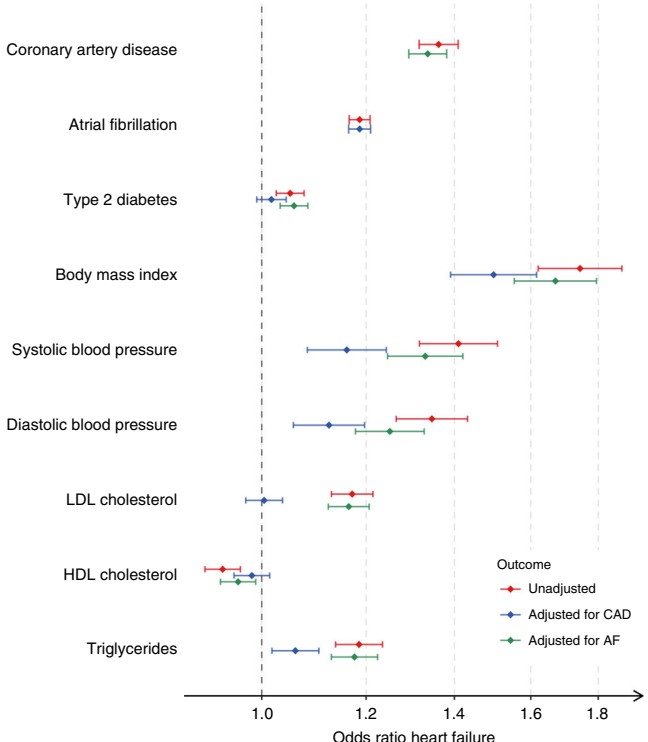

**Fig. 4 Conditional Mendelian randomisation analyses of HF risk factors.**
Forest plot of HF risk factors with significant causal effect HF risk estimated
using Mendelian randomisation, implemented with GSMR. Diamonds
represent the odds ratio and the error bars indicate the 95% confidence
interval. The unadjusted estimates represent the risk of HF as estimated
from the HF GWAS data, while the adjusted estimates represent risk of HF
conditioned, using GWAS summary statistics for atrial fibrillation (adjusted
for AF) or coronary artery disease (adjusted for CAD) estimated using the
mtCOJO method. For binary traits (coronary artery disease, atrial
fibrillation and type 2 diabetes), the MR estimates represent average causal
effect per natural-log odds increase in the trait risk. For continuous traits,
the MR estimates represent average causal effect per standard deviation
increase in the reported unit of the trait. LDL, low-density lipoprotein; HDL,
high-density lipoprotein; CAD, coronary artery disease; AF, atrial fibrillation.

Genome Sequence reference[51] or a reference sample based on 15,220 whole-
genome sequences of Icelandic individuals. The following software tools were used
by studies for phasing: Eagle[52], MaCH[53] and SHAPEIT[54]; and imputation:
mimimac2[55] and IMPUTE2[56]. For imputation to the HRC reference panel, the
Sanger Imputation Server (https://www.sanger.ac.uk/science/tools/sanger-
imputation-service) was used. The deCODE study was imputed using study specific
procedures[57]. Methods for phasing, imputation and post-imputation QC for each
study are detailed in Supplementary Data 13.

**Study-level GWA analysis.** GWA analysis for each study was performed locally
according to a common analysis plan, and summary-level estimates were provided
for meta-analysis. Autosomal single-nucleotide polymorphisms (SNPs) were tested
for association with HF using logistic regression, assuming additive genetic effects.
For the Cardiovascular Health Study, HF association estimates were generated by
analysis of incident cases using a Cox proportional hazards model. All studies
included age and sex (except for single-sex studies) as covariates in the regression
models. Principal components (PCs) were included as covariates for individual
studies as appropriate. The following tools were used for study-level GWA analysis:
ProbABEL[58], mach2dat (http://www.unc.edu/~yunmli/software.html), QuickT-
est[59], PLINK2[60], SNPTEST[61] or R[62] as detailed in Supplementary Data 13.

**QC on study summary-level data.** QC of summary-level results for each study
was performed according to the protocol described in Winkler et al.[63]. In brief, we
used the EasyQC tool to harmonise variant IDs and alleles across studies and to
compare reported allele frequencies with allele frequencies in individuals of Eur-
opean ancestry from the 1000 Genomes imputation reference panel[64]. We
inspected P–Z plots (reported P value against P value derived from the Z-score),
beta and s.e. distributions, and Manhattan plots to check for consistency and to

identify spurious associations. For each study, variants were removed if they
satisfied any one of the following criteria: imputation quality < 0.5, MAF < 0.01,
absolute betas and s.e. > 10. As recommended in Sinnott et al.[65] and Johnson
et al.[66], more stringent QC measures were applied to studies where genotyping of
cases and controls was performed on different platforms. This included more
stringent thresholds for removing SNPs with low-quality imputation, and where
available, individuals genotyped on both platforms were used to remove SNPs with
low concordance rates between the two platforms. To check for study-level
genomic inflation, we examined quantile–quantile plots and calculated the genomic
inflation factor ($\lambda_{GC}$). For three studies, where some degree of genomic inflation
was observed ($\lambda_{GC} > 1.1$), genomic control correction was applied (Supplementary
Data 13)[67].

**Meta-analysis.** Meta-analysis of summary data was conducted using the fixed-
effect IVW approach implemented in METAL (released March 25 2011)[68]. Var-
iants were included if they were present in at least half of all studies. We tested for
inflation of the meta-analysis test statistic due to cryptic population structure by
estimating the LDSC intercept, implemented using LDSC v1.0.0[13]. As the LDSC
intercept indicated no inflation (LD score intercept of 1.0069), no further correc-
tion was applied to the meta-analysis summary estimates. To identify variants
independently associated with HF, we analysed the genome-wide results using
FUMA v1.3.2[69], selecting a random sample of 10,000 UK Biobank participants of
European ancestry as an LD reference dataset[70]. Variants were filtered using a $P <
5 \times 10^{-8}$ and independent genomic loci were LD-pruned based on an $r^2 < 0.1$. We
calculated Cochrane's Q and $I^2$ statistics to assess whether the effect estimates for
HF sentinel variants were consistent across studies[71].

**Heritability estimation.** To estimate the proportion of HF risk explained by
common variants we estimated heritability $h_g^2$ on the liability scale, using LDSC on
the UK Biobank summary data (6,504 HF cases, 387,652 controls), assuming a
population prevalence of 2.5%[14]. This approach assumes that a binary trait has an
underlying continuous liability, and above a certain liability threshold an individual
becomes affected. We can then estimate the genetic contribution to the continuous
liability. Sample ascertainment can change the distribution of liability in the
sampled individuals and needs to be adjusted for, which requires making
assumptions about the population prevalence of the trait.

**LD reference dataset.** A LD reference was created, including 10,000 UK Biobank
participants of European ancestry, based on HRC-imputed genotypes (referred to
henceforth as UKB10K). European individuals were identified by projecting the UK
Biobank samples onto the 1000 G Phase 3 samples. A genomic relationship matrix
was constructed using HapMap3 variants, filtered for MAF > 0.01, $P_{HWE} < 10^{-6}$
and missingness < 0.05 in the European subset, and one member of each pair of
samples with observed genomic relatedness >0.05 was excluded to obtain a set of
unrelated European individuals. Random sampling without replacement was used
to extract a subset of 10,000 unrelated individuals of European ancestry. Variants
with a minor allele count > 5, a genotype probability > 0.9 and imputation quality
> 0.3 were converted to hard calls. This LD reference dataset was used for down-
stream summary-based analysis and for identifying SNP proxies.

**Gene set enrichment analysis.** A gene-based and gene set enrichment analysis of
variant associations was performed using MAGMA[26], implemented by FUMA
v1.3.2[69]. This analysis was performed using summary-level meta-analysis results.
First, a gene-based association analysis to identify candidate genes associated with
HF was conducted. Second, a tissue enrichment analysis of HF-associated genes
was performed using gene expression data for 30 tissues from GTEx. Finally, a gene
set enrichment analysis was performed based on pathway annotations from the
Gene Ontology database[72]. For all MAGMA analyses, multiple testing was
accounted for by Bonferroni correction.

**Missense consequences of sentinel variants and proxies.** We queried the
protein coding consequence of the sentinel variants and proxies ($r^2 > 0.8$) using the
Combined Annotation Dependent Depletion (CADD) score[73], implemented using
FUMA v1.3.2[69]. The CADD score integrates information from 63 distinct func-
tional annotations into a single quantitative score, ranging from 1 to 99, based on
variant rank relative to all 8.6 billion possible single nucleotide variants of the
human reference genome (GRCh37). Sentinel SNPs or proxies with CADD score >
20 were identified. A CADD score of 20 indicates that the variant is ranked in the
top 1% of highest scoring variants, while a CADD score of 30 indicates the variant
is ranked in the top 0.1%.

**Expression quantitative trait analysis.** To determine if HF sentinel variants had
*cis* effects on gene expression, we queried two eQTL datasets based on RNA
sequencing of human heart tissue—the GTEx v7 resource[74] and the MAGNet
repository (http://www.med.upenn.edu/magnet/). The GTExv7 sample included
272 LV and 264 RAA non-diseased tissue samples from European (83.7%) and
African Americans (15.1%) individuals. The MAGNet repository included 89 LV
and 101 LA tissue samples obtained from rejected donor tissue from hearts with no

evidence of structural disease; and 89 LV samples from individuals with DCM, obtained at the time of transplantation. eQTL analysis of the LV data from MAGNet analysis was performed using the QTLtools package[75] in DCM with adjustment for age, sex, disease status and the first three genetic PCs. To account for observed batch effects, a surrogate variant analysis was performed using the R package SVAseq[76] and 22 additional covariates were identified and included in the model. Existing eQTL summary data in LA tissue from MAGNet and heart tissue from GTEx were queried[17,77]. We queried HF sentinel variants for eQTL associations with genes located either fully or partly within a 1 megabase (Mb) region upstream or downstream of the sentinel variant (referred to as *cis*-genes). We accounted for multiple testing by adjusting a significance threshold of $P < 0.05$ for the total number of SNP-*cis*-gene tests performed across the four heart tissue eQTL datasets ($P < 4.73E-05$ for a total of 1,056 SNP–gene associations). Baseline characteristics for the MAGNet study are provided in Supplementary Table 6. We also queried sentinel HF variants for associations with *cis* gene expression in blood from the eQTLGen consortium ($N = 31,684$)[29]. Given the large sample size, we used a stringent genome-wide significance threshold of $P < 5 \times 10^{-8}$ to identify significant blood eQTLs.

**Colocalisation analysis**. Bayesian colocalisation analysis was performed using R package *coloc* to test whether shared associations with gene expression and HF risk were consistent with a single common causal variant hypothesis[30]. We tested all genes with significant *cis*–eQTL association by analysing all variants within a 200 kilobase window around the gene using eQTL summary data for heart tissues and whole blood, and HF summary data from present study. We set the prior probability of a SNP being associated only with gene expression, only with HF, or with both traits as $10^{-4}$, $10^{-4}$ and $10^{-5}$. For each gene, we report the posterior probability that the association with gene expression and HF risk is driven by a single causal variant. We consider a posterior probability of ≥0.7 as providing evidence, supporting a causal role for the gene as a mediator of HF risk.

**Transcriptome-wide association analysis**. We employed the S-PrediXcan method[78] implemented in the MetaXcan software (https://github.com/hakyimlab/MetaXcan) to identify genes whose predicted expression levels in heart tissue are associated with HF risk. Prediction models trained on GTExv7 heart tissue datasets were applied to the HERMES meta-analysis results. Only models that significantly predicted gene expression in the GTEx eQTL dataset (false discovery rate < 0.05) were considered. A total of 4859 genes were tested in left ventricle tissue and 4467 genes for right atrial appendage. Genes with an association $P < 5.36 \times 10^{-6}$ [0.05/(4859 + 4467)] were considered to have gene expression profiles significantly associated with HF.

**Protein quantitative trait analysis in blood**. We queried both *cis*- and *trans*-protein QTL (pQTL) associations based on measures for serum proteins mapping to 3000 genes in 3301 healthy individuals from the INTERVAL study[31]. We accounted for multiple testing by adjusting a significance threshold of $P < 0.05$ for the total number of tests for all variants and proteins tested (36,000 tests).

**Association of HR risk loci with other phenotypes**. We queried associations (with $P < 1 \times 10^{-5}$) of sentinel variants and proxies ($r^2 > 0.6$) with any trait in the NHGRI-EBI Catalog of published GWAS (accessed 21 January 2019)[15,79]. We report associations (where $P < 1 \times 10^{-5}$) for the sentinel variants with traits in the UK Biobank cohort using the MRBase PheWAS database (http://phewas.mrbase.org/, accessed 17 January 2019). The database contains GWA summary data for 4203 phenotypes measured in 361,194 unrelated individuals of European ancestry from the UK Biobank data. We queried GWAS data for ten traits related to HF risk factors, endophenotypes and related disease traits using summary-level data from the largest available GWAS study (either publicly available or through agreement with study investigators). The following phenotypes were considered: fractional shortening (FS), LV dimension[16], DCM; AF[17], CAD[18], LDL-C[22], T2D[23]; BMI[20], SBP and DBP[19]. For DCM, a GWAS was performed in the UKB among individuals of European ancestry with cases defined by the presence of ICD10 code I42.0 as a main/secondary diagnosis or primary/secondary cause of death with non-cases as referents, using PLINK2. Logistic regression was performed with adjustment for age, sex, genotyping array and the first ten PCs.

**Hierarchical agglomerative clustering**. We performed hierarchical agglomerative clustering on a locus level using the complete linkage method based on the associations with related traits as described above. Where a sentinel variant is not available in any of the other traits summary results, a common proxy is used in place of the sentinel variant. For the *LPA* locus, we used associations for a proxy of the more common variant (rs55730499). Dissimilarity structure was calculated using Euclidean distance based on the Z-score (beta of continuous traits or log odds of disease risk divided by s.e.) of the cross-trait associations. We accounted for multiple testing at family-wise error rate of 0.05 by Bonferroni correction for the ten traits tested per HF locus (110 tests), and considered $P < 4.5e^{-4}$ (0.05/110) as our significance threshold for association.

**Genetic correlation analysis**. We estimated genetic correlation between HF and 11 risk factors using LDSC[13] on the GWAS summary statistics for each trait: AF[17], CAD[18], LDL-C, high-density lipoprotein cholesterol (HDL-C), TGs[22], T2D[23]; BMI[20], SBP, DBP[19], HR[21] and estimated GFR[80].

**Mendelian randomisation analysis**. We performed two sample Mendelian randomisation analysis using the Generalised summary data-based Mendelian randomisation (GSMR)[25] implemented in GCTA v1.91.7beta[81]. To identify independent SNP instruments for each exposure, GWAS-significant SNPs ($P < 5 \times 10^{-08}$) for each risk factor were pruned ($r^2 < 0.05$; LD window of 10,000 kb; using the UKB10K LD reference). We then estimated the causal effect of the risk factor on the disease trait according to the MR paradigm. The HEIDI test implemented in GSMR was used to detect and remove (if HEIDI $P < 0.01$) variants showing horizontal pleiotropy i.e., having independent effects on both exposure and outcome, as such variants do not satisfy the underlying assumptions for valid instruments. As sensitivity analyses, we estimated the causal effects of known risk factors on HF risk other statistical methodology and software—the R package TwoSampleMR[82] was used to select independent variant instruments for the exposure using the same parameters as per the GSMR analysis ($P < 5 \times 10^{-8}$; $r^2 < 0.05$; LD window of 10,000 kb), except the TwoSampleMR package uses the 1000 Genomes as the LD reference. Causal estimates based on the IVW[83], MR-Egger and median-weighted methods[84] were then calculated using the Mendelian Randomisation[85] R package. To enable comparison of MR estimates between traits, we present effect estimates corresponding to the risk of HF for a 1-s.d. higher risk factor of interest. Where the original GWAS conducted rank-based inverse normal transformation (RINT) of a trait prior to GWAS, we used the per-allele beta coefficients following RINT to approximate the equivalent values on the standardised scale, as has been conducted previously.

To determine if the causal effects of the continuous risk factors on HF were mediated via their effects on CAD or AF risk, we repeated the GSMR analysis after conditioning the HF summary statistics on CAD and AF GWAS summary statistics, as described below.

**Conditional analysis**. To estimate the effects of HF risk variants after adjusting for risk factors which showed a significant causal effect on HF in the MR analyses, we performed the mtCOJO on summary data, as implemented in GCTA v1.91.7beta[81]. HF summary statistics were adjusted for AF[17], CAD[18], LDL-C, HDL-C, TGs[22], DBP, SBP[19] and BMI[20] using GWAS summary data. The UKB10K LD reference was used.

**Reporting summary**. Further information is provided in the Nature Research Reporting Summary.

## Data availability
The datasets generated during this study are available from the corresponding author upon reasonable request. The summary GWAS estimates for this analysis are available on the Cardiovascular Disease Knowledge Portal (http://www.broadcvdi.org/).

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

## Acknowledgements

We acknowledge the contribution from the EchoGen Consortium. A full list of contributing authors and further acknowledgements are given in Supplementary Notes 4 and 5.

## Author contributions

S. Shah, J.B.W., F.A., A.D.H., C.C.L., J.G.S., R.S.V., D.I.S. and R.T.L. are members of HERMES executive committee. S. Shah, A. Henry, H. Holm, M.V.H., F.A., A.D.H., K. Kuchenbaecker, P.T.E., C.C.L., J.G.S., R.S.V., D.I.S. and R.T.L. drafted and finalised the manuscript. S. Shah, A. Henry, C.R., H.L., G.S., Å.K.H., M.D.C., C.A., W.C., S.D., D.F.G., P.v.d.H., E.I., R.C.L., T.M., C.P.N., T.N., B.M.P., K.M.R., S.P.R.R., J.v.S., N.L.S., P. Svensson, K.D.T., G.T., B.T., A.A.V., X.W., H.X., H. Hemingway, N.J.S., J.J.M., J.Y., P.M.V., A. Malarstig, H. Holm, S.A.L., N.S., M.V.H., T.P.C., F.A., A.D.H., K. Kuchenbaecker, P.T.E., C.C.L., J.G.S., R.S.V., D.I.S. and R.T.L. contributed to and revised the manuscript. C.R., H.L., G.S., G.F., Å.K.H., J.B.W., M.P.M., M.D.C., A. Helgadottir, N.V., A.D., P.A., C.A., K.G.A., J.Ä., J.D.B., M.L.B., H.L.B., J.B., Broad AF Investigators, M.R.B., L.B., D.J.C., R.G.C., D.I.C., Xing Chen, Xu Chen, J.C., J.P.C., G.E.D., S.D., A.S.D., M.D., S.C.D., M.E.D., EchoGen Consortium, G.E., T.E., S.B.F., C.F., I.F., M.G., S. Ghasemi, V.G., F.G., J.S.G., S. Gross, D.F.G., R.G., C.M.H., P.v.d.H., C.L.H., E.I., J.W.J., M.K., K. Khaw, M.E.K., L.K., A.K., C.L., L.L., C.M.L., B.L., L.A.L., J.L., P.M., A. Mahajan, K.B.M., W.M., O.M., I.R.M., A.D.M., A.P.M., A.C.M., M.W.N., C.P.N., A.N., T.N., M.L.O., A.T.O., C.N.A.P., H.M.P., M.P., E.P., B.M.P., K.M.R., P.M.R., S.P.R.R., J.I.R., P. Salo, V.S., A.A.S., D.T.S., N.L.S., S. Stender, D.J.S., P. Svensson, M. Tammesoo, K.D.T., M. Teder-Laving, A.T., G.T., U.T., C.T., S.T., A.G.U., A.V., U.V., A.A.V., N.J.W., D.W., P.E.W., R.W., K.L.W., L.M.Y., B.Y., F.Z., J.H.Z., N.J.S., C.N., A. Malarstig, H. Holm, S.A.L., N.S., T.P.C., K. Kuchenbaecker, P.T.E., C.C.L., K.S., J.G.S., R.S.V., D.I.S. and R.T.L. contributed to study-specific GWAS by providing phenotype data or performing data analyses. S. Shah and H.L. performed meta-analyses. C.R., M.P.M., J.B., K.B.M. and T.P.C. provided heart eQTL data, and contributed to analysis. S. Shah, A. Henry, C.R., G.F., M.V.H. and R.T.L. performed downstream analyses. S. Shah, F.A., A.D.H., K. Kuchenbaecker, P.T.E., C.C.L., J.G.S., R.S.V., D.I.S. and R.T.L. conceived, designed, and supervised the overall project. Contribution statements from Regeneron Genetics Center are provided in Supplementary Note 6. All authors have approved the final version of the manuscript.

## Competing interests

J.B.W., L.B., Xing Chen, C.L.H., M.W.N. and A. Malarstig are current or former employee of Pfizer who may hold Pfizer stock and/or stock options. J.D.B. and J.C. are employees of Regeneron Genetics Center. M.E.D. is an employee of Regeneron Pharmaceuticals. W.M. reports grants and personal fees from Siemens Diagnostics, grants and personal fees from Aegerion Pharmaceuticals, grants and personal fees from AMGEN, grants and personal fees from Astrazeneca, grants and personal fees from Danone Research, personal fees from Hoffmann LaRoche, personal fees from MSD, grants and personal fees from Pfizer, personal fees from Sanofi, personal fees from Synageva, grants and personal fees from BASF, grants from Abbott Diagnostics, grants and personal fees from Numares AG, grants and personal fees from Berlin-Chemie, employment with Synlab Holding Deutschland GmbH, all outside the submitted work. M.L.O. reports grant support from GlaxoSmithKline, Eisai, Janssen, Merck and AstraZeneca. B.M.P. serves on the DSMB of a clinical trial funded by Zoll LifeCor and on the Steering Committee of the Yale Open Data Access Project funded by Johnson & Johnson. V.S. participated in a conference trip sponsored by Novo Nordisk and received a honorarium from the same source for participating in an advisory board meeting. He also has ongoing research collaboration with Bayer Ltd. B.T. is a full-time employee of Servier. S.A.L. receives sponsored research support from Bristol Myers Squibb/Pfizer, Bayer AG and Boehringer Ingelheim, and has consulted for Abbott, Quest Diagnostics and Bristol Myers Squibb/Pfizer. M.V.H. has collaborated with Boehringer Ingelheim in research, and in accordance with the policy of the The Clinical Trial Service Unit and Epidemiological Studies Unit (University of Oxford), did not accept any personal payment. P.T.E. receives sponsored research support from Bayer AG, and has consulted with Bayer AG, Novartis and Quest Diagnostics. D.I.S. is a full-time employee of BenevolentAI. R.T.L. has received research grants from Pfizer. The remaining authors declare no competing interest.

## Additional information

Sonia Shah [1,2,3,112], Albert Henry [2,3,4,112], Carolina Roselli [5,6], Honghuang Lin [7,8], Garðar Sveinbjörnsson [9], Ghazaleh Fatemifar [3,4,10], Åsa K. Hedman [11], Jemma B. Wilk [12], Michael P. Morley [13], Mark D. Chaffin [5], Anna Helgadottir [9], Niek Verweij [5,6], Abbas Dehghan [14,15], Peter Almgren [16], Charlotte Andersson [8,17], Krishna G. Aragam [5,18,19], Johan Ärnlöv [20,21], Joshua D. Backman [22], Mary L. Biggs [23,24], Heather L. Bloom [25], Jeffrey Brandimarto [13], Michael R. Brown [26], Leonard Buckbinder [12], David J. Carey [27], Daniel I. Chasman [28,29], Xing Chen [12], Xu Chen [30], Jonathan Chung [22], William Chutkow [31], James P. Cook [32], Graciela E. Delgado [33], Spiros Denaxas [3,4,10,34,35], Alexander S. Doney [36], Marcus Dörr [37,38], Samuel C. Dudley [39], Michael E. Dunn [40], Gunnar Engström [16], Tõnu Esko [5,41], Stephan B. Felix [37,38], Chris Finan [2,3], Ian Ford [42], Mohsen Ghanbari [43],

Sahar Ghasemi[38,44], Vilmantas Giedraitis [45], Franco Giulianini[28], John S. Gottdiener[46], Stefan Gross [37,38], Daníel F. Guðbjartsson [9,47], Rebecca Gutmann[48], Christopher M. Haggerty[27], Pim van der Harst[6,49,50], Craig L. Hyde[12], Erik Ingelsson [51,52,53,54], J. Wouter Jukema [55,56], Maryam Kavousi[43], Kay-Tee Khaw[57], Marcus E. Kleber [33], Lars Køber[58], Andrea Koekemoer [59], Claudia Langenberg [60], Lars Lind[61], Cecilia M. Lindgren [5,62,63], Barry London[64], Luca A. Lotta[60], Ruth C. Lovering [2,3], Jian'an Luan [60], Patrik Magnusson [30], Anubha Mahajan [63], Kenneth B. Margulies[13], Winfried März[32,65,66], Olle Melander[67], Ify R. Mordi [36], Thomas Morgan[31,68], Andrew D. Morris [69], Andrew P. Morris[32,63], Alanna C. Morrison[26], Michael W. Nagle[12], Christopher P. Nelson [59], Alexander Niessner[70], Teemu Niiranen[71,72], Michelle L. O'Donoghue[73], Anjali T. Owens[13], Colin N.A. Palmer [36], Helen M. Parry[36], Markus Perola[71], Eliana Portilla-Fernandez[43,74], Bruce M. Psaty[75,76], Regeneron Genetics Center, Kenneth M. Rice [23], Paul M. Ridker[28,29], Simon P.R. Romaine[59], Jerome I. Rotter [77], Perttu Salo[71], Veikko Salomaa [71], Jessica van Setten [78], Alaa A. Shalaby[79], Diane T. Smelser[27], Nicholas L. Smith[76,80,81], Steen Stender[82], David J. Stott[83], Per Svensson [84,85], Mari-Liis Tammesoo[41], Kent D. Taylor [86], Maris Teder-Laving [41], Alexander Teumer [38,44], Guðmundur Thorgeirsson[9,87], Unnur Thorsteinsdottir[9,88], Christian Torp-Pedersen[89,90,91], Stella Trompet[55,92], Benoit Tyl [93], Andre G. Uitterlinden [43,94], Abirami Veluchamy[36], Uwe Völker [38,95], Adriaan A. Voors[7], Xiaosong Wang[31], Nicholas J. Wareham[60], Dawn Waterworth[96], Peter E. Weeke[58], Raul Weiss[97], Kerri L. Wiggins [24], Heming Xing[31], Laura M. Yerges-Armstrong[96], Bing Yu[26], Faiez Zannad[98], Jing Hua Zhao[60], Harry Hemingway [3,4,10,99], Nilesh J. Samani[59], John J.V. McMurray[99], Jian Yang [1,100], Peter M. Visscher [1,100], Christopher Newton-Cheh[5,19,101], Anders Malarstig[11,12], Hilma Holm[9], Steven A. Lubitz[5,102], Naveed Sattar [99], Michael V. Holmes[103,104,105], Thomas P. Cappola[13], Folkert W. Asselbergs [2,3,78], Aroon D. Hingorani[2,3], Karoline Kuchenbaecker [106,107], Patrick T. Ellinor [5,102], Chim C. Lang[36], Kari Stefansson[9,88], J. Gustav Smith[5,108,109], Ramachandran S. Vasan [8,110], Daniel I. Swerdlow[2] & R. Thomas Lumbers [3,4,10,111]*

[1]Institute for Molecular Bioscience, The University of Queensland, Brisbane, Queensland 4072, Australia. [2]Institute of Cardiovascular Science, University College London, London, UK. [3]British Heart Foundation Research Accelerator, University College London, London, UK. [4]Institute of Health Informatics, University College London, London, UK. [5]Program in Medical and Population Genetics, The Broad Institute of MIT and Harvard, Cambridge, MA, USA. [6]Department of Cardiology, University Medical Center Groningen, University of Groningen, Groningen, The Netherlands. [7]Section of Computational Biomedicine, Department of Medicine, Boston University School of Medicine, Boston, MA, USA. [8]National Heart, Lung, and Blood Institute's and Boston University's Framingham Heart Study, Framingham, MA, USA. [9]deCODE genetics/Amgen Inc., Sturlugata 8, 101, Reykjavik, Iceland. [10]Health Data Research UK London, University College London, London, UK. [11]Cardiovascular Medicine unit, Department of Medicine Solna, Karolinska Institute, Stockholm, Sweden. [12]Pfizer Worldwide Research & Development, 1 Portland St, Cambridge, MA, USA. [13]Penn Cardiovascular Institute, Perelman School of Medicine, University of Pennsylvania, Philadelphia, PA, USA. [14]Department of Epidemiology and Biostatistics, Imperial College London, St Mary's Campus, London W2 1PG, UK. [15]MRC-PHE Centre for Environment and Health, Department of Epidemiology and Biostatistics, Imperial College London, St Mary's Campus, London W2 1PG, UK. [16]Department of Clinical Sciences, Lund University, Malmö, Sweden. [17]Department of Cardiology, Herlev Gentofte Hospital, Herlev Ringvej 57, 2650 Herlev, Denmark. [18]Center for Genomic Medicine, Massachusetts General Hospital, Boston, MA, USA. [19]Cardiovascular Research Center, Massachusetts General Hospital, Boston, MA, USA. [20]Department of Neurobiology, Care Sciences and Society/ Section of Family Medicine and Primary Care, Karolinska Institutet, Stockholm, Sweden. [21]School of Health and Social Sciences, Dalarna University, Falun, Sweden. [22]Regeneron Genetics Center, 777 Old Saw Mill River Road, Tarrytown, NY 10591, USA. [23]Department of Biostatistics, University of Washington, Seattle, WA, USA. [24]Department of Medicine, University of Washington, Seattle, WA, USA. [25]Division of Cardiology, Department of Medicine, Emory University Medical Center, Atlanta, GA, USA. [26]Department of Epidemiology, Human Genetics, and Environmental Sciences, The University of Texas School of Public Health, Houston, Texas, USA. [27]Department of Molecular and Functional Genomics, Geisinger, Danville, PA, USA. [28]Division of Preventive Medicine, Brigham and Women's Hospital, Boston, MA 02215, USA. [29]Harvard Medical School, Boston, MA 02115, USA. [30]Department of Medical Epidemiology and Biostatistics, Karolinska Institutet, Stockholm, Sweden. [31]Novartis Institutes for Biomedical Research, Cambridge, MA, USA. [32]Department of Biostatistics, University of Liverpool, Liverpool, UK. [33]Vth Department of Medicine (Nephrology, Hypertensiology, Endocrinology, Diabetology, Rheumatology), Medical Faculty of Mannheim, University of Heidelberg, Heidelberg, Germany. [34]The National Institute for Health Research University College London Hospitals Biomedical Research Centre, University College London, London, UK. [35]The Alan Turing Institute, London, United Kingdom. [36]Division of Molecular & Clinical Medicine, University of Dundee, Ninewells Hospital and Medical School, Dundee, DD1 9SY, UK. [37]Department of Internal Medicine B, University Medicine Greifswald, Greifswald, Germany. [38]DZHK (German Center for Cardiovascular Research), partner site Greifswald, Greifswald, Germany. [39]Cardiovascular Division, Department of Medicine, University of Minnesota, Minneapolis, MN, USA. [40]Regeneron Pharmaceuticals, Cardiovascular Research, 777 Old Saw Mill River Road, Tarrytown, NY 10591, USA. [41]Estonian Genome Center, Institute of Genomics, University of Tartu, Tartu 51010, Estonia. [42]Robertson Center for Biostatistics, University of Glasgow, Glasgow, UK.

[43]Department of Epidemiology, Erasmus University Medical Center, Rotterdam, The Netherlands. [44]Institute for Community Medicine, University Medicine Greifswald, Greifswald, Germany. [45]Department of Public Health and Caring Sciences, Geriatrics, Uppsala University, Uppsala 75185, Sweden. [46]Department of Medicine, Division of Cardiology, University of Maryland School of Medicine, Baltimore, MD, USA. [47]School of Engineering and Natural Sciences, University of Iceland, 101 Reykjavik, Iceland. [48]Division of Cardiovascular Medicine, University of Iowa Carver College of Medicine, Iowa City, IA, USA. [49]Department of Genetics, University Medical Center Groningen, University of Groningen, Groningen, The Netherlands. [50]Durrer Center for Cardiogenetic Research, ICIN-Netherlands Heart Institute, Utrecht, The Netherlands. [51]Department of Medicine, Division of Cardiovascular Medicine, Stanford University School of Medicine, Stanford, CA 94305, USA. [52]Stanford Cardiovascular Institute, Stanford University, Stanford, CA 94305, USA. [53]Department of Medical Sciences, Molecular Epidemiology and Science for Life Laboratory, Uppsala University, Uppsala, Sweden. [54]Stanford Diabetes Research Center, Stanford University, Stanford, CA 94305, USA. [55]Department of Cardiology, Leiden University Medical Center, Leiden, The Netherlands. [56]Einthoven Laboratory for Experimental Vascular Medicine, LUMC, Leiden, The Netherlands. [57]Department of Public Health and Primary Care, University of Cambridge, Cambridge CB2 0QQ, UK. [58]Department of Cardiology, Copenhagen University Hospital Rigshospitalet, Copenhagen, Denmark. [59]Department of Cardiovascular Sciences, University of Leicester and NIHR Leicester Biomedical Research Centre, Glenfield Hospital, Leicester, UK. [60]MRC Epidemiology Unit, Institute of Metabolic Science, University of Cambridge School of Clinical Medicine, Cambridge CB2 0QQ, UK. [61]Department of Medical Sciences, Uppsala University, Uppsala, Sweden. [62]Big Data Institute at the Li Ka Shing Centre for Health Information and Discovery, University of Oxford, Oxford, UK. [63]Wellcome Trust Centre for Human Genetics, University of Oxford, Oxford, UK. [64]Division of Cardiovascular Medicine and Abboud Cardiovascular Research Center, University of Iowa, Iowa City, IA, USA. [65]Synlab Academy, Synlab Holding Deutschland GmbH, Mannheim, Germany. [66]Clinical Institute of Medical and Chemical Laboratory Diagnostics, Medical University of Graz, Graz, Austria. [67]Department of Internal Medicine, Clinical Sciences, Lund University and Skåne University Hospital, Malmö, Sweden. [68]Vanderbilt University School of Medicine, Nashville, TN, USA. [69]Usher Institute of Population Health Sciences and Informatics, University of Edinburgh, Edinburgh, United Kingdom. [70]Department of Internal Medicine II, Division of Cardiology, Medical University of Vienna, Vienna, Austria. [71]National Institute for Health and Welfare, Helsinki, Finland. [72]Department of Medicine, Turku University Hospital and University of Turku, Turku, Finland. [73]TIMI Study Group, Cardiovascular Division, Brigham and Women's Hospital, Boston, MA, USA. [74]Division of Vascular Medicine and Pharmacology, Department of Internal Medicine, Erasmus University Medical Center, Rotterdam, The Netherlands. [75]Department of Medicine, Epidemiology, and Health Services, University of Washington, Seattle, WA, USA. [76]Kaiser Permanente Washington Health Research Institute, Kaiser Permanente Washington, Seattle, WA, USA. [77]The Institute for Translational Genomics and Population Sciences, Departments of Pediatrics and Medicine, Los Angeles Biomedical Research Institute at Harbor-UCLA Medical Center, Torrance, CA, USA. [78]Department of Cardiology, Division Heart and Lungs, University Medical Center Utrecht, University of Utrecht, Utrecht, The Netherlands. [79]Division of Cardiology, Department of Medicine, University of Pittsburgh Medical Center and VA Pittsburgh HCS, Pittsburgh, PA, USA. [80]Department of Epidemiology, University of Washington, Seattle, WA, USA. [81]Seattle Epidemiologic Research and Information Center, Department of Veterans Affairs Office of Research & Development, Seattle, WA, USA. [82]Department of Clinical Biochemistry, Copenhagen University Hospital, Herlev and Gentofte, København, Denmark. [83]Institute of Cardiovascular and Medical Sciences, College of Medical, Veterinary and Life Sciences, University of Glasgow, Glasgow, United Kingdom. [84]Department of Clinical Science and Education, Södersjukhuset, Karolinska Institutet, Stockholm, Sweden. [85]Department of Cardiology, Södersjukhuset, Stockholm, Sweden. [86]Institute for Translational Genomics and Population Sciences, LABiomed and Departments of Pediatrics at Harbor-UCLA Medical Center, Torrance, CA 90502, USA. [87]Division of Cardiology, Department of Internal Medicine, Landspitali, National University Hospital of Iceland, Hringbraut, 101 Reykjavik, Iceland. [88]Faculty of Medicine, Department of Medicine, University of Iceland, Saemundargata 2, 101, Reykjavik, Iceland. [89]Department of Epidemiology and Biostatistics, Aalborg University Hospital, Aalborg, Denmark. [90]Department of Health, Science and Technology, Aalborg University Hospital, Aalborg, Denmark. [91]Departments of Cardiology, Aalborg University Hospital, Aalborg, Denmark. [92]Section of Gerontology and Geriatrics, Department of Internal Medicine, Leiden University Medical Center, Leiden, The Netherlands. [93]Translational and Clinical Research, Servier Cardiovascular Center for Therapeutic Innovation, 50 rue Carnot, 92284 Suresnes, France. [94]Department of Internal Medicine, Erasmus MC, University Medical Center Rotterdam, Rotterdam, The Netherlands. [95]Interfaculty Institute for Genetics and Functional Genomics, University Medicine Greifswald, Greifswald, Germany. [96]Human Genetics, GlaxoSmithKline, Collegeville, PA, USA. [97]Division of Cardiovascular Medicine, Department of Internal Medicine, The Ohio State University Medical Center, Columbus, OH, USA. [98]Université de Lorraine, CHU de Nancy, Inserm and INI-CRCT (F-CRIN), Institut Lorrain du Coeur et des Vaisseaux, 54500 Vandoeuvre Lès, Nancy, France. [99]BHF Cardiovascular Research Centre, University of Glasgow, Glasgow, United Kingdom. [100]Queensland Brain Institute, The University of Queensland, Brisbane, QLD 4072, Australia. [101]Center for Human Genetic Research, Massachusetts General Hospital, Boston, MA, USA. [102]Cardiac Arrhythmia Service and Cardiovascular Research Center, Massachusetts General Hospital, Boston, MA, USA. [103]Medical Research Council Population Health Research Unit at the University of Oxford, Oxford, UK. [104]Clinical Trial Service Unit and Epidemiological Studies Unit, Nuffield Department of Population Health, Big Data Institute, University of Oxford, Oxford, UK. [105]National Institute for Health Research Oxford Biomedical Research Centre, Oxford University Hospital, Oxford, UK. [106]Division of Psychiatry, University College of London, London W1T 7NF, UK. [107]UCL Genetics Institute, University College London, London WC1E 6BT, UK. [108]Department of Cardiology, Clinical Sciences, Lund University and Skåne University Hospital, Lund, Sweden. [109]Wallenberg Center for Molecular Medicine and Lund University Diabetes Center, Lund University, Lund, Sweden. [110]Sections of Cardiology, Preventive Medicine and Epidemiology, Department of Medicine, Boston University Schools of Medicine and Public Health, Boston, MA, USA. [111]Bart's Heart Centre, St. Bartholomew's Hospital, London, UK. [112]These authors contributed equally: Sonia Shah, Albert Henry. A full list of consortium members appears at the end of the paper. *email: t.lumbers@ucl.ac.uk

## Regeneron Genetics Center

Goncalo Abecasis[22], Joshua Backman[22], Xiaodong Bai[22], Suganthi Balasubramanian[22], Nilanjana Banerjee[22], Aris Baras[22], Leland Barnard[22], Christina Beechert[22], Andrew Blumenfeld[22], Michael Cantor[22], Yating Chai[22], Jonathan Chung[22], Giovanni Coppola[22], Amy Damask[22], Frederick Dewey[22], Aris Economides[22], Gisu Eom[22], Caitlin Forsythe[22], Erin D. Fuller[22], Zhenhua Gu[22], Lauren Gurski[22], Paloma M. Guzzardo[22], Lukas Habegger[22], Young Hahn[22], Alicia Hawes[22], Cristopher van Hout[22], Marcus B. Jones[22], Shareef Khalid[22], Michael Lattari[22], Alexander Li[22], Nan Lin[22], Daren Liu[22], Alexander Lopez[22], Kia Manoochehri[22], Jonathan Marchini[22],

Anthony Marcketta[22], Evan K. Maxwell[22], Shane McCarthy[22], Lyndon J. Mitnaul[22], Colm O'Dushlaine[22], John D. Overton[22], Maria Sotiropoulos Padilla[22], Charles Paulding[22], John Penn[22], Manasi Pradhan[22], Jeffrey G. Reid[22], Thomas D. Schleicher[22], Claudia Schurmann[22], Alan Shuldiner[22], Jeffrey C. Staples[22], Dylan Sun[22], Karina Toledo[22], Ricardo H. Ulloa[22], Louis Widom[22], Sarah E. Wolf[22], Ashish Yadav[22] & Bin Ye[22]

