## [Peer Review File · Nature Communications]

Reviewers' Comments:

Reviewer #1:

Remarks to the Author:

I am still concerned that the data sources likely carry a strong selection bias for specific disease entities. For example, cohorts that were exclusively selected from cardiac catheterization laboratories will have an overrepresentation of CAD cases (LURIC). Population based samples will have a stronger representation of patients with arterial hypertension etc.. This limitation should be further explored by stratification and discussed more thoroughly. Moreover, the authors should provide more details on individual data of the participants rather than describing crudely their sources in the appendix.

I am also doubtful that such diverse samples are suitable for MR analyses and thus support the far reaching conclusions in this respect. For example, if both cases with HF and controls without HF have had a myocardial infarction (EPHESUS; SOLID) then the association study will fail to detect CAD risk alleles. If most cases with HF and controls without HF have diabetes (Go-Darts) then the association study will fail to detect diabetes risk alleles. Accordingly, any sort of adjustment or MR study will underestimate the effects of respective risk alleles!

A major aim of genetic association studies in complex disorders is to learn more about the molecular aetiology. This reviewer still suggests in addition to a strategy of pooling all individuals irrespectively of the disease causing condition to study individuals in whom HF is secondary to a well-defined condition, e.g. dilated cardiomyopathy or myocardial infarction, separately.

Table 1 should mention for each of the loci the established genomewide significant associations with conditions that predispose the heart failure (CAD, obesity/BMI, aFib, hypertension). This increases the information content and puts the data into perspective.

Table 1, it is misleading to call the 9p21 locus by a gene that has been shown to be not involved in the aetiology. One way of dealing with the locus could be to name it 9p21/CDKN2B.

The lambda for the meta-analysis was now reported to be 1.127. This is fairly high and it might be good to have a genetic epidemiologist comment on this issue.

The authors should discuss the point that the signals they picked up for HF represent, not surprisingly, the strongest in the respective class for CAD, aFIB and BMI.

Abstract and Line 349 ff and Line 485/Conclusion: Given that the case-control samples are not representative (some include only CAD patients +/- HF) conditioned HF GWAS summary statistics using mtCOJO does not definitively exclude a role of respective risk factors as intermediary phenotypes. The authors should be much more explicit in this respect – and more conservative regarding their conclusion that most risk loci shared by CAD and HF are not related to CAD in the first place.

Line 391 ff: The authors should better clarify why they restricted this analysis to myocardial tissue – and report what is known from previous studies on the respective loci (e.g. LPA).

The authors may want to pay attention to a recent study linking the genetics of CAD with HF (J Am Coll Cardiol. 2019 Jun 18;73(23):2932-2942.).

Line 345 ff: Six sentinel variants 345 were also associated with CAD, including established loci

such as CDKN2B-AS1(9p21) and LPA2. The authors should drop “also”, because their own data show that HF is likely to be a consequence of the association with CAD.

Reviewer #2:

Remarks to the Author:

The authors have responded in a comprehensive manner to my previous comments. I agree that remaining questions e.g. association by HF subtype can be the topic of future analyses.

Reviewers' comments:

Reviewer #1 (Remarks to the Author):

I am still concerned that the data sources likely carry a strong selection bias for specific disease entities. For example, cohorts that were exclusively selected from cardiac catheterization laboratories will have an overrepresentation of CAD cases (LURIC). Population based samples will have a stronger representation of patients with arterial hypertension etc.. This limitation should be further explored by stratification and discussed more thoroughly. Moreover, the authors should provide more details on individual data of the participants rather than describing crudely their sources in the appendix.

We agree with the reviewer that non-population cohorts differ with respect to upstream or co-morbid disease phenotypes. Detailed information on the demographic and clinical characteristics of participating studies are given in Supplementary Table 16. Of the 26 included studies, 9 were from non-population samples, accounting for ~18% of the total case population, of which 5 were performed in a uniform risk population (CAD – EPHEBUS, SOLID; suspected CAD – LURIC; diabetes – GoDARTS; elevated cardiovascular risk – PROSPER). To highlight this, we have added the following text to the results section (**Main text 309-311**):

*The study sample comprised both population cohorts (17 studies, 38,780 HF cases, 893,657 controls) and case-control samples (9 studies, 8,529 cases, 36,357 controls); (see **Supplementary Note** for a detailed description of the included studies).*

We provide empirical evidence to show that inclusion of such studies does not materially influence the estimates derived from Mendelian randomization (please see response to following reviewer question).

We acknowledge the importance of stratified analysis for analysis of this complex phenotype and this will form the basis for our next collaborative meta-analysis. In this study we will perform large-scale stratified analysis using harmonised covariates for stratification. We highlight this future work in the discussion (**Main text 509-513**).

I am also doubtful that such diverse samples are suitable for MR analyses and thus support the far reaching conclusions in this respect. For example, if both cases with HF and controls without HF have had a myocardial infarction (EPHEBUS; SOLID) then the association study will fail to detect CAD risk alleles. If most cases with HF and controls without HF have diabetes (Go-Darts) then the association study will fail to detect diabetes risk alleles. Accordingly, any sort of adjustment or MR study will underestimate the effects of respective risk alleles!

We agree that the effects of upstream risk factor-associated alleles on heart failure will be underestimated in HF case-control studies performed within populations recruited

with the corresponding risk background. Similarly, MR analysis would underestimate the overall effects of the given risk factor on HF.

To estimate the possible effects from the inclusion of these studies in the meta-analysis (~18% of cases, ~4% controls), we have undertaken a sensitivity analysis by including only population samples in the meta-analysis (17 studies, 38,780 HF cases, 893,657 controls). We found that the effect estimates for both the HF-associated risk factor loci and two-sample Mendelian randomisation analysis were consistent with the results from the full sample. The exclusion of case-control studies performed in patients with or at risk of CAD did not reduce the effect estimates of the MR analysis for CAD or for those HF risk loci with established CAD associations.

Trait	GSMR (HF based on all studies) N=47,309 HF cases				GSMR (HF based on population cohorts only) N=38,780 HF cases			
	Beta	SE	P-value	Nsnps	Beta	SE	P-value	Nsnps
Body Mass Index	0.556	0.0373	2.67E-50	78	0.566	0.0397	2.76E-46	79
Diastolic Blood Pressure	0.0263	0.00281	9.13E-21	111	0.0291	0.00302	5.98E-22	111
Glomerular filtration rate	0.26	0.148	0.08	54	0.238	0.159	0.13	54
Heart Rate	-0.00219	0.0025	0.38	97	-0.00336	0.00269	0.21	97
High Density Lipoprotein	-0.0682	0.0158	1.58E-05	144	-0.0569	0.0169	0.00077	150
Low Density Lipoprotein	0.158	0.0185	1.11E-17	126	0.151	0.0195	9.06E-15	131
Systolic Blood Pressure	0.0166	0.00168	4.82E-23	100	0.0192	0.0018	1.40E-26	100
Triglycerides	0.17	0.0209	3.80E-16	105	0.166	0.0223	1.25E-13	105
Atrial fibrillation	0.171	0.00928	1.40E-75	147	0.171	0.01	3.71E-65	146
Coronary artery disease	0.309	0.0174	1.67E-70	43	0.3	0.0184	7.81E-60	44
Type 2 diabetes	0.0497	0.0124	6.35E-05	47	0.0526	0.0133	7.76E-05	47

We have added the following text to summarise the results of these analyses (**Main text lines 463-467**):

We then performed a sensitivity analysis to explore potential bias arising from the inclusion of case-control samples by repeating the Mendelian randomisation analysis using heart failure GWAS estimates generated from population cohort studies only. The results of this analysis were consistent with those generated from the overall sample (data not shown).

A major aim of genetic association studies in complex disorders is to learn more about the molecular aetiology. This reviewer still suggests in addition to a strategy of pooling all individuals irrespectively of the disease causing condition to study individuals in whom HF is secondary to a well-defined

condition, e.g. dilated cardiomyopathy or myocardial infarction, separately.

We agree that looking at both pooled and stratified samples is important as reflected in previous responses and in the main text discussion. These analyses are the focus of our next large-scale collaborative effort. For the purposes of this study, we designed and validated interoperable clinical classifiers against adjudicated cases populations to harmonise phenotypes across studies and to achieve sufficient statistical power.

Table 1 should mention for each of the loci the established genomewide significant associations with conditions that predispose the heart failure (CAD, obesity/BMI, aFib, hypertension). This increases the information content and puts the data into prospective.

We provide the association of sentinel variants or proxies with heart failure related traits (including those suggested) in Supplementary Table 4. To make the table clearer we have filtered the results to present only those associations that reach genome-wide significance ($P < 5 \times 10^{-8}$). We are happy to add this information to Table 1 however as an alternative, given limited space, we have coded this information into **Figure 3** (see below).

Table 1, it is misleading to call the 9p21 locus by a gene that has been shown to be not involved in the aetiology. One way of dealing with the locus could be to name it 9p21/CDKN2B.

We thank the reviewer for their suggestion and have now amended the manuscript to refer to this locus as 9p21/CDKN2B.

The lambda for the meta-analysis was now reported to be 1.127. This is fairly high and it might be good to have a genetic epidemiologist comment on this issue.

The LD score regression intercept has been shown to provide a more powerful and accurate correction factor than genomic control (Bulik-Sullivan et al Nat Genet. 2015; PMID 25642630) and was recommended by co-authors who are statistical geneticists (including Jian Yang and Peter Visscher). This approach has also been used in several recent GWAS publications to show no inflation due to population structure despite high lambda GC e.g. Watson et al Nat Gen 2019 (lambda GC 1.22; LD intercept 1.02) and Howard et al Nat Neuroscience 2019 (lambda GC 1.63; LD intercept 1.015)

The authors should discuss the point that the signals they picked up for HF represent, not surprisingly, the strongest in the respective class for CAD, aFIB and BMI.

We agree and have updated the discussion as follows (**Main text 490-492**):

The identified loci were associated with modifiable risk factors and traits related to LV structure and function and include the strongest associations signals from GWAS of CAD (9p21, LPA), AF (PITX2), and BMI (FTO).

Abstract and Line 349 ff and Line 485/Conclusion: Given that the case-control samples are not representative (some include only CAD patients +/- HF) conditioned HF GWAS summary statistics using mtCOJO does not definitively exclude a role of respective risk factors as intermediary phenotypes. The authors should be much more explicit in this respect – and more conservative regarding their conclusion that most risk loci shared by CAD and HF are not related to CAD in the first place.

Thank you for this suggestion. We have added text to highlight this potential limitation. We have performed a sensitivity analysis using summary data generated from only population cohorts and find no change in the MR estimates (**main text 464-467**).

Line 391 ff: The authors should better clarify why they restricted this analysis to myocardial tissue – and report what is known from previous studies on the respective loci (e.g. LPA).

Given the modest sample size of eQTL datasets, it is unlikely we have the power to test all genes in multiple different tissues. We therefore used a data driven approach to determine the most relevant tissue for heart failure and complemented this with an analysis of whole blood for which a very large sample was available (n = 31, 684).

We have clarified our rationale by adding the following text (**main text 392-402**):

We then sought to identify candidate genes for HF risk loci by assessing their effects on gene expression. Given that cardiac dysfunction defines HF and that HF-associated genes by MAGMA analysis were enriched in heart tissues, we first looked for expression quantitative trait loci (eQTL) in heart tissues (LV, left atrium, and right atrium auricular region) from the Myocardial Applied Genomics Network (MAGNet) and Genotype-Tissue Expression (GTEx) projects. Three of 12 variants were significantly associated with the expression of one or more genes located in cis in at least one heart tissue (Bonferroni-

corrected $P < 0.05$) (Supplementary Table 8). For several of the identified HF loci, extra-cardiac tissues are likely to be relevant; for example, liver is reported to mediate effects of the LPA locus (doi: 10.1101/518290). To begin to explore these effects, we analysed results from a large whole blood eQTL dataset ($n = 31,684$) and found associations with cis gene expression ($P < 5 \times 10^{-8}$) for 8 of 12 sentinel variants (Supplementary Table 9)

The authors may want to pay attention to a recent study linking the genetics of CAD with HF (J Am Coll Cardiol. 2019 Jun 18;73(23):2932-2942.).

Thank you for alerting us to this interesting study. We have now referenced this study in the main text (**line 454**).

Line 345 ff: Six sentinel variants 345 were also associated with CAD, including established loci such as CDKN2B-AS1(9p21) and LPA2. The authors should drop “also”, because their own data show that HF is likely to be a consequence of the association with CAD.

We have edited the text to remove “also” on line 345 (line 349 in latest version).

Reviewers' Comments:

Reviewer #2:

Remarks to the Author:

No further comments.